# iPSC-Derived Cardiomyocytes in Inherited Cardiac Arrhythmias: Pathomechanistic Discovery and Drug Development

**DOI:** 10.3390/biomedicines11020334

**Published:** 2023-01-25

**Authors:** Eline Simons, Bart Loeys, Maaike Alaerts

**Affiliations:** Cardiogenetics Research Group, Center of Medical Genetics, University of Antwerp and Antwerp University Hospital, 2650 Antwerp, Belgium

**Keywords:** iPSC, cardiac arrhythmia, iPSC-derived cardiomyocytes

## Abstract

With the discovery of induced pluripotent stem cell (iPSCs) a wide range of cell types, including iPSC-derived cardiomyocytes (iPSC-CM), can now be generated from an unlimited source of somatic cells. These iPSC-CM are used for different purposes such as disease modelling, drug discovery, cardiotoxicity testing and personalised medicine. The 2D iPSC-CM models have shown promising results, but they are known to be more immature compared to in vivo adult cardiomyocytes. Novel approaches to create 3D models with the possible addition of other (cardiac) cell types are being developed. This will not only improve the maturity of the cells, but also leads to more physiologically relevant models that more closely resemble the human heart. In this review, we focus on the progress in the modelling of inherited cardiac arrhythmias in both 2D and 3D and on the use of these models in therapy development and drug testing.

## 1. Introduction

Since the discovery of induced pluripotent stem cells (iPSCs) in 2006 by Takahashi and Yamanaka [1], iPSCs have increasingly gained popularity in the scientific field; not only to perform stem cell research but also to create somatic cells derived from these iPSCs such as neurons [2], cardiomyocytes [3] and hepatic cells [4] amongst many others. The numerous advantages, such as access to difficult-to-access human cell types, the development of patient-specific cell types, decreased need for laboratory animals and less ethical concerns compared to embryonal stem cells (ESC), are well-known. However, there are also some drawbacks on the use of these derived cells such as variability, low differentiation efficiency and the immature state of the differentiated cells. Nevertheless, iPSC-derived cells are indispensable in the current cell-biology research community.

In 2009, Zwi et al. presented their work on the development of a way to differentiate iPSCs into cardiomyocytes [3]. Their iPSC-derived cardiomyocytes (iPSC-CM) expressed the cardiac specific markers cardiac troponin-I and sarcomeric α-actinin, were electrophysiologically active and they displayed the expected response to the admission of different drugs. Ever since, an increasing number of papers have been published using iPSC-CM to model diseases, perform drug and cardiotoxicity testing and develop new therapies.

In this review, we take a closer look at these recent developments focusing on cardiac arrhythmia disorders and the transition from 2D to 3D culture models (Figure 1).

## 2. iPSC-Derived Cardiomyocytes as Inherited Cardiac Arrhythmia Models

Inherited cardiac arrhythmias are characterised by the dysfunction of cardiac ion channels, their accessory proteins or cell–cell contact proteins which can lead to ventricular arrhythmias and potential sudden cardiac death. The most well-known inherited cardiac arrhythmias include long QT syndrome (LQTS), Brugada syndrome (BrS), catecholaminergic polymorphic ventricular tachycardia (CPVT), short QT syndrome (SQTS) and arrhythmogenic cardiomyopathy (ACM). These diseases are caused by pathogenic variants in genes encoding components or accessory elements of these ion channels or desmosomes. The type of mutation (loss-of-function (LOF) or gain-of-function (GOF)) is also important in defining the disease outcome.

The arrhythmias mostly occur in the ventricles, making ventricular cardiomyocytes the most relevant cell type to investigate. Most of the currently used iPSC-CM differentiation protocols generate a mixture of atrial, ventricular and sinoatrial pacemaker cardiomyocytes, but with a clear overrepresentation/higher presence of ventricular cells. Ventricular action potentials (AP) are characterised by a more negative maximum diastolic potential, a rapid AP upstroke, a long plateau phase and an APD90/APD50 ratio ≤1.3/1.4 [5,6,7]. It is also possible to differentiate iPSCs directly into the specific cardiomyocyte types [8].

In 2018, Garg et al. reviewed the published iPSC-CM models of several channelopathies [9] and Pan et al. updated this review with the addition of ACM (Appendix A) [10]. Here, the overview is updated (see Table 1) with more recently published models.

### 2.1. Long QT Syndrome

Long QT syndrome has a prevalence of 1 in 2000 and is clinically diagnosed by a prolongation of the QT interval (heart rate-corrected QT interval ≥480 ms) on the electrocardiogram (ECG) [31]. Currently, there are 17 subtypes of LQTS based on the gene involved and the most common subtypes are LQT1, LQT2 and LQT3, caused by mutations in the *KCNQ1*, *KCNH2* and *SCN5A* genes, respectively [32]. Over the years, several LQTS iPSC-CM models have been published, the first in 2010 by Moretti et al. [6]. The latter investigated patient-specific iPSC-CM of three related LQT1 patients harbouring a p.(Arg190Gln) variant and showed a prolonged action potential duration at 90% of repolarisation (APD90) and lower potassium current densities compared to control individuals. This corresponded to the phenotype observed in the patients. Since then, several papers have been published describing LQTS iPSC-CM models of known pathogenic mutations (reviewed by Garg et al. [9], Appendix A). More recently, LQTS iPSC-CM models have been used to investigate the pathogenicity of variants of uncertain significance (VUS). For example, Garg et al. created a LQT2 iPSC-CM model harbouring the VUS p.(Thr983Ile) in the *KCNH2* gene. Using CRISPR/Cas9 technology, they developed both a homozygous VUS cell line as well as an isogenic control line. Both patch-clamp and multi-electrode array (MEA) experiments showed prolonged APD50, APD90 and field potential duration (FPD) in the homozygous as well as in the heterozygous VUS iPSC-CMs. In addition, more beat irregularity or early after depolarisations (EADs) were observed and the phenotype of the homozygous VUS iPSC-CMs resembled that of the known pathogenic p.(Ala561Val) *KCNH2* variant. Potassium (I_Kr_) current was decreased in the VUS cell line and restored to normal current densities in the isogenic control [11]. Chavali et al. took a different CRISPR/Cas9 approach when they introduced a VUS p.(Asn639Thr) in the *CACNA1C* gene into iPSCs to create a patient-independent iPSC model. Prolonged APD and FPD were recorded due to a slower inactivation of the Ca_v_1.2 current. As this cellular phenotype recapitulated the patient phenotype, the authors reclassified the VUS as probably pathogenic [12].

### 2.2. Brugada Syndrome

Brugada syndrome is a cardiac arrhythmia with a prevalence ranging from 1 in 500 to 1 in 2000 and patients display a specific ST-segment elevation on the ECG. Many genes encoding ion channels and accessory proteins have been associated with the disease but only one is currently considered as causal, namely *SCN5A,* encoding the cardiac sodium channel Na_V_1.5. Mutations in *SCN5A* account for up to 20–25% of the BrS cases [33,34]. The first report on iPSC-CM in BrS was published by Davis et al. They modelled an iPSC-CM line harbouring a *SCN5A* mutation p.(1798insAsp) from a patient with an overlap syndrome of LQT/BrS and conduction disorder. Reduced and persistent sodium currents, slower upstroke velocity and prolongation of APD90 were observed in patients’ iPSC-CMs but not in controls, mimicking the LOF and GOF phenotype of this mutation [13]. Later, two iPSC-CM lines from BrS patients with *SCN5A* (p.(Arg620His)+ p.(Arg811His) and c. 4190delA) mutation were evaluated by Liang et al. Both cell lines showed abnormal action potentials (AP) compared to the controls as well as a reduced sodium current [14]. In 2021, Nijak et al. published a review on iPSC-CM models generated of BrS patients, included in Appendix A [35]. More recently, extra reports on BrS iPSC-CM models harbouring variants in *SCN5A* (p.(Val1405Met), p.(Ser1812X))*, SCN1B* (p.(Ala197Val)) and *CACNB2* (p.(Ser142Phe)) were published. Reduced expression of the encoded proteins was observed as well as reduced sodium or calcium currents leading to reduced action potential amplitude (APA) and maximum upstroke velocity (Vmax) but prolonged APDs [15,16,17]. Calcium imaging showed more proarrhythmic events such as EADs and DADs (early and delayed after depolarisations) in BrS cell lines compared to control cell lines [16]. A *SCN5A* p.(Ser1812X) variant resulted in reduced conduction velocity and proarrhythmic events [17].

### 2.3. Short QT Syndrome

Short QT syndrome is diagnosed by a shortening of the QT interval on the ECG and has a prevalence ranging from 1 in 1000 to 1 in 5000 [36]. Causal GOF mutations are mostly found in potassium channel genes such as *KCNH2*, *KCNQ1* and *KCNJ2* [37]. The first iPSC-CM model of SQTS was published by El-Battrawy et al. in 2018 where one patient cell line with a p.(Asn588Lys) mutation in the *KCNH2* gene was compared to two control cell lines. They demonstrated an upregulation of the hERG channel expression and increased potassium currents (I_Kr_) resulting in a shortening of the action potential. During calcium-handling experiments, irregular beating, DAD-like and EAD-like arrhythmic events were recorded more in patient iPSC-CMs compared to control iPSC-CMs [18]. Later, the same mutation and another *KCNH2* p.(Thr618Ile) variant were modelled in iPSC-CMs and similar electrophysiological and molecular results were obtained [19,20]. An iPSC-based cardiac cell sheet model was created by Shinnawi and colleagues and an increase in susceptibility to the development of re-entrant arrhythmias recorded [19]. The p.(Thr618Ile) variant did not give rise to any arrhythmic events. However, there was an increased beat-to-beat variability in the patient cell line [20].

### 2.4. Catecholaminergic Polymorphic Ventricular Tachycardia Type

CPVT most often occurs in young adults and athletes and is triggered by β-adrenergic stimulation related to exercise or emotional stress. It is mainly caused by mutations in Ca^2+^-handling related genes such as *RYR2* and *CASQ2* and has an estimated prevalence of 1 in 10,000 [37,38]. Both proteins are essential in the Ca^2+^ handling in heart and muscle cells, responsible for the proper contraction of the cells. Different iPSC-CM CPVT models have been developed (reviewed by Garg et al. in 2018 [9], included in Appendix A). The first CPVT iPSC-CM model from a patient carrying a *RYR2* pathogenic variant (p.Phe2483Ile) was published in 2011 by Fatima et al. The analysis revealed more DAD events in patient iPSC-CMs compared to control iPSC-CMs and embryonal stem cell-derived cardiomyocytes (ESC-CM), recapitulating the CPVT phenotype. The underlying aberrant sarcoplasmic reticulum (SR) Ca^2+^ release in the iPSC-CMs is responsible for the development of these DADs and arrhythmias [21]. The same variant was modelled using CRISPR/Cas9 by Wei et al. and showed longer calcium sparks in both hetero- and homozygous iPSC-CMs, larger SR Ca^2+^ leak levels and smaller load levels which is consistent with higher diastolic Ca^2+^ levels [22]. In 2018, Acimovic et al. published an iPSC-CM model of a CPVT patient with a *RYR2* p.(Asp3638Ala) variant. They found an increase in beat rate in the patient cell line compared to both iPSC- and ESC-derived CMs and a weaker response in force contraction upon stress induction. Calcium handling was normal under basal conditions, but upon stress more irregular Ca^2+^-release events in CPVT iPSC-CMs were recorded. Patch clamp data revealed a prolongation of the AP in basal conditions while during stress, APD, Vmax and the amplitude were lower in CPVT CMs compared to controls [23]. Several other reports on *RYR2* variants, either from patient-specific [24,26] or CRIPSR/Cas9-induced iPSC-CMs [25], show similar aberrant Ca^2+^ handling although mutant lines also differ from each other, for example in the magnitude of the Ca^2+^ leak or SR Ca^2+^ content [24,25,26]. Two *CASQ2* (p.(Asp307His)) patient-specific iPSC-CM models showed DADs, oscillatory prepotentials, after-contractions and diastolic [Ca^2+^]_i_ rises similar to *RYR2* CPVT models [27].

### 2.5. Arrhythmogenic Cardiomyopathy

ACM, previously known as arrhythmogenic right ventricular cardiomyopathy (ARVC), is a rare disease (1 in 5000) that is characterised by fibrofatty myocardial replacement, leading to impaired ventricular systolic function and ventricular arrhythmias. Mutations in desmosomal genes such as *PKP2, DSG2, DSP, DSC2* and *JUP* play a prominent role in the development of the disease [39]. The first model of ARVC was published in 2013 by Ma et al. They created a patient-specific iPSC-CM model with a *PKP2* p.(Leu614Pro) mutation and showed downregulation of the expression of plakophilin and plakoglobin but no other desmosomal genes [28]. El-Battrawy and Buljubasic studied the same patient-derived iPSC-CM ACM model harbouring a mutation in the *DSG2* gene (p.(Gly638Arg)) [29,30]. The amplitude and the upstroke velocity of the AP were decreased as well as peak I_Na_, I_NCX_, I_to_, I_SK_ and I_KATP_, while I_Kr_ on the contrary was enhanced. Mutant iPSC-CMs showed more arrhythmogenic effects compared to control cells [29]. In addition, Buljubasic further investigated the underlying molecular mechanisms and revealed upregulation of SK4 channels and NDPK-B resulting in increased I_SK4_, pacemaker activity and arrhythmic events [30].

## 3. From 2D to 3D

In the iPSC-CM field, immaturity of the created iPSC-CM is a well-known problem. As the cardiomyocytes often only stay in culture for 30 days or less, it is not surprising that the phenotype of these cells does not fully recapitulate the phenotype of a mature native cardiomyocyte that has been developing for many years. Ahmed et al. (2020) reviewed the currently applied methods of maturation and pinpointed the main differences between fetal-like iPSC-CMs and adult cardiomyocytes. Methods to promote maturation include prolonged culture, addition of hormones (e.g., thyroid hormone) or cellular energy source (e.g., fatty acids such as palmitate, oleic acid, linoleic acid), co-culture, extracellular matrix, mechanical or electrical stimulation and 3D culture [40]. The latter is not only beneficial for the maturity of the cardiomyocytes but also enables the creation of 3D models that are more similar to native heart tissue. The heart consists of cardiomyocytes, but also various other cell types are present in the tissue such as endothelial cells (EC), fibroblasts (FB), pericytes, smooth muscle cells, immune cells (myeloid and lymphoid), adipocytes, mesothelial cells and neuronal cells [41]. Meanwhile, Pinto et al. found that CMs accounted for only 25–35% of the cells in the heart, ECs for 60% and FBs for less than 20%; Litviňuková found CMs represented 30% to 50% of the cells in atrial and ventricular samples, respectively, while ECs represented 10% and FBs 20% [41,42]. Adding these extra cell types to the model will make it even more physiologically relevant and likely more suitable for modelling pathological conditions and downstream applications such as drug or cardiotoxicity screening.

Below, we will discuss the development from 2D to 3D iPSC-CM cultures with or without other cell types using scaffold-free and scaffold-based techniques.

### 3.1. Scaffold-Free 3D Culture

One method to create a 3D cell culture is the scaffold-free hanging droplet method in which iPSC-CMs are placed in a droplet in an ultra-low attachment plate with [43,44] or without [45,46] the addition of other cell types such as cardiac fibroblasts and endothelial cells. Beauchamp et al. and Ergir et al. reported a long-term stable 3D model of iPSC-CMs that was able to respond to electrical, pharmacological, and physical stimuli but Ca^2+^ dyes only partially penetrated the culture and the CMs still displayed more fetal-like features such as shorter sarcomeres [45,46]. Sharma et al. combined iPSC-CM with human cardiac fibroblasts (HCFs) and human coronary artery endothelial cells to create cardiac spheroids containing a cardiac endothelial cell network that recapitulated better than the in vivo human heart [44].

Organoids are mainly formed by differentiating iPSC directly to CM (and other cell types) in ultra-low attachment plates. Drakhlis et al. generated a model of heart-forming organoids (HFO) by differentiating free-floating iPSC aggregates into cardiac organoids that resemble the early embryonic heart as they are composed of a myocardial layer and endocardial-like cells. They were able to model a *NKX2*.5 KO which resulted in similar cardiac malformations such as decreased cardiomyocyte adhesion and hypertrophy as observed in in vivo mouse studies [47]. A similar HFO protocol by Lewis-Israeli et al. using different small molecules’ concentrations and adding one WNT pathway modulation step enabled the generation of multiple cardiac-specific cell lineages such as endo- and epicardial cells, endothelial cells and cardiac fibroblasts [48]. Lee at al. started from embryonic bodies and generated chamber-forming HFOs. RNA-seq revealed that they more closely resembled the fetal heart than adult heart tissue, but here as well, several cell types were generated [49]. As such, a drawback of this technique is that the iPSC-CMs still display an immature phenotype but the HFOs are well suited to studying cardiac diseases linked to development.

Another scaffold-free method is used to create cardiac microtissues (cMT) where several (previously generated) cell types (CMs, ECs, FBs, …) are combined. Giacomelli et al. combined iPSC-derived ECs, iPSC-derived cardiac FBs and iPSC-CMs to form a microtissue displaying mature iPSC-CM ultrastructures such as elongated tubular myofibrils and T-tubule-like structures [50]. RNA-seq indicated a mature expression profile of the iPSC-CMs comparable to that of adult CMs. Electrophysiological maturation was proven by the presence of the typical AP notch, although this has also been observed in 2D cultures [17,51]. As a proof-of-concept they created a cMT consisting of healthy iPSC-CMs and iPSC-ECs combined with mutant cardiac FB of an ACM patient with a *PKP2* (c.2013delC, p.(Lys672ArgfsX12)) mutation (Table 2) and found reduced Cx43 expression in ACM cMT as well as arrhythmic behaviour [50], highlighting the importance of the presence of these non-myocytes in the model. In another paper, a LQTS cMT harbouring a *KCNQ1* p.(Arg594Gln) variant, showed a prolonged field potential compared to wild-type cMT [52] proving that the cMT can recapitulate the disease phenotype (Table 2). However, as 2D models already showed this phenotype, the MT model was not of specific added value in this case.

### 3.2. Scaffold-Based 3D Culture

Another frequently used method is scaffold-based culture. These scaffolds consist of (decellularised) extracellular matrix (ECM) [54], natural or synthetic polymers [55,56] and can be combined as a hydrogel in an organised well-defined shape or in certain orientations [57]. Fong et al. tested the effect of adult and fetal extracellular matrix from decellularised bovine adult and fetal heart tissue on the maturity of the CMs in both 2D and 3D cultures. Adult heart ECM improved maturation, demonstrated by increased expression of several calcium-handling genes and enhanced calcium signalling, both in 2D and 3D culture with the highest expression levels observed in 3D cultured iPSC-CMs. However, there was no improvement on the formation of T-tubules [54].

In engineered heart tissue (EHT) iPSC-CMs are grown on hydrogel scaffolds wrapped around two flexible pillars that have the ability to mechanically stimulate the cells and improve maturation. Several published models indeed prove that CMs grown in EHT present more mature electrophysiological properties such as action potential amplitude and upstroke velocity and more mature rod-shape morphology and sarcomere alignment [58]. Expression profiles as well as the cardiac ultrastructure, bioenergetics and t-tubule formation of stimulated EHT are more in line with adult cardiac tissue then fetal cardiac tissue [59]. To improve maturation even more, Lu et al. induced progressive stretch on the EHT which led to higher contractility and passive elasticity, more mature excitation/contraction coupling and a higher ratio of beta-myosin heavy chain (MHC) by alpha-MHC mRNA [60]. Goldfracht et al. combined the use of ECM with EHT, and in comparison, using a 2D model they found an increased expression of cardiac-related genes and the cardiomyocytes were arranged anisotropically and developed relatively elongated and oriented cell alignments. They created a LQTS2 (*KCNH2* p.(Ala614Val)) and CPVT2 (*CASQ2* p.(Asp307His)) (Table 2) model and using voltage and calcium dyes, AP prolongation in LQTS iPSC-CM was revealed while the CPVT cell model showed abnormal calcium transients and more arrhythmias under stress conditions, indicating that these EHT models can be used to study channelopathies. In comparison with a 2D single cell model, the EHT showed less frequent, severe or complicated arrhythmogenic activity which is clinically more relevant as the extremely high incidence of arrhythmias as recorded in a single cell model would probably be incompatible with life. Re-entrant arrhythmias were not observed at baseline in the LQT-EHT but they were developed after blocking the I_Kr_, mimicking the clinical situation in LQT patients challenged with a QT prolonging agent [53]. The major advantage of this technique is the maturation state of the CMs, but special equipment for the generation of this EHT is needed, which might not be available for every lab.

### 3.3. Heart-on-a-Chip

Heart-on-a-chip is a method to culture iPSC-CM—with or without other cell types —in a 2D or 3D manner on a microfluidic device in a chamber with built-in channels for fluids, microactuators and microsensors [61]. Microactuators can give either electrical or mechanical stimuli to the cells/tissue, while the sensors record electrophysiological signals or contraction force [61]. Heart-on-a-chip has been used for drug toxicity assessments and maturation was shown to be improved through electrical and mechanical stimulation [62]. Although some cardiac disease models such as ischaemia and fibrosis have been investigated using the heart-on-a-chip method [63,64], to date there are no publications on its use for inherited cardiac arrhythmias. The technique is currently still under development and the primary focus is on its application for drug cardiotoxicity screening. Even for this application, there are some challenges such as standardisation, reliable tissue manufacturing, high throughput, high content functional readouts and high cost, that still need to be solved before heart-on-a-chip can be more widely used [65,66].

## 4. Drug and Gene Therapy Testing

### 4.1. Cardiotoxicity Screening

A first application of iPSC-CMs and their ability to model/display/show arrhythmias and structural pathology is testing of the cardiotoxicity of a drug under development. Cardiotoxicity and arrhythmia induction such as life-threatening Torsade de pointes (TdP) are a main reason for preclinical and clinical drug failure and withdrawal from the market. In 2013, the Comprehensive in Vitro Proarrhythmia Assay (CiPA) initiative was founded to overcome the low specificity of the preclinical studies and clinical trials at that time [67]. One of the novel components is testing the effect of a drug in vitro in iPSC-CM. A total of 28 compounds with known cardiac effects were tested in commercially available iPSC-CMs using a MEA system and voltage-sensitive dyes and could be classified as high-, intermediate- and low-risk for TdP [68]. To confirm these findings, these drugs were tested over several laboratories/facilities, commercial cardiomyocyte types and different MEA platforms and reproducible concentration-dependent electrophysiological responses were reported, indicating that iPSC-CMs can predict clinical QT prolongation and/or arrhythmogenic potential of drug compounds [69,70,71]. Lee at al showed that addition of a contractility assay (impedance measurement) into the evaluation of cardiotoxicity provides/allows more mechanistic insights on the drug effect [72]. As discussed above, 3D heart-on-a-chip models are also being tested, holding promise for even better prediction of cardiotoxic and pro-arrhythmic drug effects as they better recapitulated the clinical effects compared to 2D iPSC-CM models as they present occasionally with arrhythmias that are not reported in adult cardiomyocytes [73,74]. Regarding inherited cardiac arrhythmias, variable expressivity is a known characteristic, with many individuals who carry pathogenic variants remaining asymptomatic throughout life. However, specific drugs can also elicit life-threatening arrhythmias in these carriers/patients and patients are recommended to avoid taking them. Using iPSC-CM with such pathogenic variants in cardiotoxicity screening could be a valuable option to predict these adverse effects in a subset of the population.

### 4.2. Drug Testing

In addition to cardiotoxicity, iPSC-CM can also be deployed to test compounds that could (partially) restore the phenotype of inherited cardiac arrhythmias models (Table 3). Two recent publications reported a 2D LQT3 (*SCN5A* p.(Phe1473Cys)) model that was used to test mexiletine and different analogues in their ability to reduce the prolongation of the AP and they found that the analogues were more potent and selective in inhibiting the late sodium current, responsible for the APD prolongation in patients. In addition, they did not induce AP prolongation or EADs, known off-target effects of mexiletine due to unwanted inhibition of hERG [75], and were still able to suppress arrhythmias [76,77]. Verapamil and lidocaine were able to reduced APD in another LQT model harbouring to variants (*KCNQ1* p.(Gly219Glu)/*TRPM4* p.(Thr160Met)) [78].

Several LQT2 models, with pathogenic variants in the *KCNH2* (hERG channel) have also been used to test drugs. Telmisartan and GW0742 are agonists of the PPARδ pathway, which helps hERG to stabilise the PKA-phosphorylated active state of the channel opening at more negative potentials. Duncan et al. tested these agonists in a patient iPSC-CM model harbouring a *KCNH2* p.(Ala561Thr) variant and found a 20% reduction in APD for both compounds, which is comparable to the observed effect of NS1643 (also 20% APD shortening), a known compound that reduces inactivation of the hERG channel [79]. Mehta et al. created iPSC-CMs of five patients with either disrupted *KCNH2* trafficking (p.(Ala561Val), (IVS9-28A/G)) or synthesis (p.(Ser428X), p.(Arg366X)) to test the use of lumacaftor as a treatment option as the drug acts as a chaperone during protein folding. As predicted, they found higher *KCNH2* expression and shortened field potentials after 7 days of treatment with lumacaftor in patients with trafficking defect mutations but not in patients with disrupted synthesis of the hERG channel [80]. Two of the patients received treatment with lumacaftor and Ivacaftor and indeed showed a shorter QTc, however this shortening was not as pronounced as in the in vitro model indicating that the translation from in vitro to in vivo is not straightforward [90]. Another study also tested lumacaftor on three LQT2 (*KCNH2*) patient iPSC-CM lines with different pathogenic variants and found rescued phenotypes in two (p.(Asn633Ser), p.(Arg685Pro)) of the three lines. For the third one (p.(Gly604Ser)), on the other hand, they saw a prolongation of the AP after administration of the compound, which could be explained by the dominant-negative effect that was observed next to the trafficking defect caused by the third variant [81]. Another compound (ICA-105574, a type II I_Kr_ activator) was used by two groups and tested both on VUS (p.(Thr983Ile)) and pathogenic (p.(Ala422Thr) LQT2 iPSC-CM models. They both saw a shortening of the action potential, field potential or calcium transient but with the risk of overcorrection at higher concentrations which might induce arrhythmic events [11,82].

Ajmaline is a class IA anti-arrhythmic drug that can be used to diagnose BrS patients. Studies have already shown that ajmaline can inhibit various currents, including I_Na_, I_to_ or I_Kr_ [83]. In the iPSC-CM of a BrS patient without a known genetic cause ajmaline had the same blocking effect on both the repolarisation and depolarisation caused by an inhibition of both I_Na_ and I_Kr_ as observed in the control iPSC-CMs. In an iPSC-CM model harbouring two *SCN10A* (p.(Arg1268Gln)/p.(Arg1250Gln) variants there was a more pronounced reduction in APA and Vmax compared to control iPSC-CMs [84]. The same was observed in a *SCN1B* (p.(Leu210Pro)/p.(Pro213Thr)) iPSC-CM model [85]. Cilostazol and milrinone, two phosphodiesterase III inhibitors, increased I_Ca_ and suppressed I_to_ by increasing the heart rate [91]. These were tested on BrS iPSC-CM models from two patients carrying a *SCN5A* p.(Ser1812X) variant, which resulted in a reduction in I_to_ and arrhythmic beating [17]. Bisoprolol, a beta blocker, was recently tested in a *CACNB2* p.(Ser142Phe) iPSC-CM model and reduced variation in beat-to-beat interval time as well as arrhythmic events. Quinidine, a class I antiarrhythmic agent, on the other hand, only reduced arrhythmic events [15]. The same anti-arrhythmic effect of quinidine was observed in a *SCN5A* (p.(Val1405Met)) and *SCN1B* (p.(Ala197Val)) iPSC-CM model [16].

Guo et al. tested quinidine in an iPSC-CM model of a SQT (*KCNH2*, p.(Thr618Ile)) patient who was already receiving quinidine treatment. The cell model confirmed the beneficial effect of quinidine as APD was prolonged, comparable to the APD of the isogenic control. Next to quinidine, a short peptide derived from a scorpion, BmKKx2, prolonged the APD by targeting the *KCNH2* gene [20]. In another model (*KCNH2* p.(Asn588Lys)), quinidine reduced Vmax, prolonged APD and abolished arrhythmic events while sotalol and metoprolol did not have an effect [18]. Ivabradine, ajmaline, and mexiletine prolonged APD and reduced arrhythmic events in the same iPSC-CM model [86].

One way to prevent arrhythmias in CPVT is to upregulate the calcium uptake by the mitochondria by, for example, mitochondrial Ca^2+^ uptake enhancers (MiCUp) such as efsevin and kaempferol [87]. These MiCUps were tested both in mice and *RYR2* (p. (Ser406Leu)) patient iPSC-CMs and were able to reduce episodes of stress-induced ventricular tachycardia in mice and reduce arrhythmogenic Ca^2+^ waves in iPSC-CMs [87]. Two other MiCUps, ezetimibe and disulfiram, suppressed arrhythmogenesis in patient iPSC-CMs [88] (genetic variant not specified). Another way to modulate calcium is by inhibiting the Ca^2+^/calmodulin-dependent protein kinase II (CaMKII) with a CaMKII inhibitory peptide, which is successful in reducing the abnormal Ca^2+^ release events and frequency of Ca^2+^ sparks in two CPVT *RYR2* (p.(Ser404Arg)/p.(Asn658Ser), p.(Gly3946Ser)/(p.(Gly1885Glu)) iPSC-CMs [89]. EL20, a tetracaine derivative and *RYR2* inhibitor, decreased spark activity in iPSC-CMs of a CPVT patient harbouring a *RYR2* (p.(Arg176Gln)) mutation without negatively affecting the Ca^2+^ transient amplitude [24]. Stutzman et al. created four iPSC-CM lines of CPVT patients with *RYR2* mutations, (p.(Phe13Leu), p.(Leu14Pro), p.(Arg15Pro) and p.(Arg176Gln)), and treated them with nadolol and flecainide. Both were able to decrease the Ca^2+^ transient amplitude and spark activity [26].

All these reports confirm the great potential of iPSC-CM arrhythmia models to test novel and existing therapies, and also for personalised medicine. In both 2D and 3D, they could also be effectively used for larger drug-library screening experiments.

### 4.3. Gene Therapy Testing

Inherited cardiac arrhythmia iPSC-CM models have also been used to test novel gene therapies, acting straight on the nucleic acid molecular/genetic level.

One way to perform gene therapy is by patient-specific targeting the causal mutation. Matsa et al. used an allele-specific small interfering RNA to knock down the mutated *KCNH2* mRNA in LQTS (*KCNH2* p.(Ala561Thr)) patient iPSC-CMs thereby preventing the dominant negative-trafficking defect. This resulted in a shortening of the AP, increase in K^+^ current and rescue of the arrhythmogenic phenotype [92]. A more general gene therapy approach was published by Dotzler et al. They developed a novel method with a dual mode of action called suppression-and-replacement (SupRep) *KCNQ1* gene therapy. As the name indicates, first the endogenous alleles were suppressed by short hairpin RNA (shRNA) and in the next step, the *KCNQ1* gene was replaced by expression of a shRNA-immune (shIMM) *KCNQ1* cDNA immune for breakdown by the shRNA. This method was tested in four LQT1 (*KCNQ1* p.(Tyr171X), p.(Val254Met), p.(Ile567Ser) and p.(Ala344Ala/splice variant)) patient iPSC-CM models and showed a shortening of the APD in all 2D patient models. As a proof-of-concept, a 3D cardiac organoid of one of the patient lines (p.(Tyr171X)) was created and here as well, an APD shortening was observed after treatment [93]. The same treatment approach was used for *KCNH2* variants, in iPSC-CM models of two LQT2 (p.(Gly604Ser), p.(Asn633Ser)) patients as well as in one SQT (p.(Asn588Lys) patient and resulted in a normal APD90 for both the LQT2 and SQT patients [94].

## 5. Discussion and Conclusions

With the advent of iPSC creation, major steps have been taken to differentiate these stem cells into several cell types including iPSC-derived cardiomyocytes. Using this model in inherited cardiac arrhythmia research has increased knowledge on the underlying disease mechanisms and creates opportunities to functionally characterise and interpret the pathogenicity of patient-specific genetic variants and to perform (personalised) drug testing. As a proof-of-concept of this more ‘personalised’ drug testing, a few ‘clinical trials in a dish’ have been performed where healthy control individuals and their iPSC-CMs were challenged with known QT-prolonging drugs to compare the effect on the in vitro model to the in vivo situation. Using sotalol, a correlation was found between the in vivo QT interval and in vitro FPD results [95]. One study also found such a positive correlation for moxifloxacin [96] while another did not find a correlation between the APD response slopes and clinical QT response to moxifloxacin or dofetilide [97]. Using (subject-specific) iPSCs for research and drug testing also requires the use of a comprehensive informed consent explaining future use of created iPSCs and derivatives. The reported 2D iPSC-CM disease models recapitulate the patients’ phenotype at the cellular level, however, if the specific tested characteristics are compared over several iPSC-clones or several different papers, quite some variability can be observed [98]. In addition, for example, the iPSC-CM models of BrS patients with an unknown genetic cause did not show any electrophysiological differences compared to healthy control iPSC-CMs [99]. The known immature phenotype of iPSC-CMs with immature ion channel expression most likely plays a role in these observations and small changes in ionic currents might not be picked up. More in-depth analysis of the iPSC-CM cellular disease phenotype including transcriptomics or proteomics approaches could be useful to further characterise these models.

In addition, efforts have been made to improve the maturity of iPSC-CMs, with one important strategy to culture them in 3D models such as microtissues, organoids and engineered heart tissue. Amongst others, Kerr et al. showed that iPSC-CM in 3D cultures showed a higher similarity to human adult myocardial transcriptome compared to 2D models and had enhanced cell–cell communication, ECM organisation and vascularisation capacity [100]. The addition of other (iPSC-derived) cell types that are present in heart tissue further improves the physiological relevance and maturation state of the model. Use of these 3D models will certainly increase the suitability for disease modelling and drug testing. It should be taken into account, though, that they are more complex at the culture level—complicating the high-throughput needed for larger screenings, so that extra variability is introduced to an already variable model [98] and the complexity of the analysis is also increased. Light for microscopy, fluorescent dyes and drugs need to penetrate deeper and evenly into the 3D culture to reach all cells, more computational power might be needed and more expensive single-cell analysis approaches such as scRNA-seq could be necessary. Indeed, Feng et al. already performed single cell analysis on cardiac organoids and found more differentially expressed genes in iPSC-CMs compared to other cell types present in the organoid between Ebstein’s anomaly patients and healthy controls [101].

Despite the immense progress that has been made in iPSC-CM generation and application potential, some limiting factors such as immaturity, genetic and phenotypic heterogeneity and variability still have an impact on their usability and should be kept in mind when translating the results in vivo [98]. For clinical application in regenerative medicine the arrhythmogenic potential, immunogenicity, tumorigenicity and heterogeneity of the iPSC-CMs should be taken into consideration. In conclusion, iPSC-CMs have been instrumental in modelling inherited cardiac arrhythmias, small-scale testing of disease-specific drugs or gene therapies and cardiotoxicity testing. The transition from 2D to 3D models has improved cellular maturity and physiological relevance, but also increases the complexity of the model and its analysis. Large-scale drug library screenings have not yet been performed, but further automation and high-throughput analysis methods will certainly pave the way for this application. Further evolution of both 2D and 3D iPSC-CM modelling and analysis techniques will allow the discovery of new treatment options for cardiac arrhythmias in general as well as for personalised medicine.

## Figures and Tables

**Figure 1 biomedicines-11-00334-f001:**
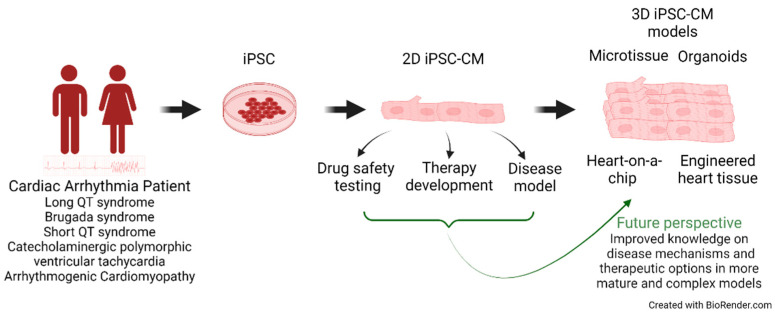
Schematic representation of different applications of iPSC-cardiomyocytes.

**Table 1 biomedicines-11-00334-t001:** Overview of published 2D iPSC-CM cardiac arrhythmia disease models.

Syndrome	Causal Gene Variant	Experimental Approach	Cellular Phenotype	Ref.
LQTS	*KCNQ1* p.(Arg190Gln)	PC, IF	Prolonged AP, reduced I_Ks_ current, ER retention, increased susceptibility to catecholamine-induced tachyarrhythmia, attenuation of this phenotype with beta blockade	[6]
	*KCNH2* p.(Thr983Ile)	PC, MEA, WB, CI	Prolonged APD50 and APD90, beat irregularity, EAD, decreased I_Kr_ density, reduced channel surface expression, higher diastolic Ca^2+^	[11]
	*CACNA1C* p.(Asn639Thr)	CardioExcyte 96, PC	Prolonged Maximum Field Potential Duration and APD, slower Ca_V_1.2 voltage-dependent inactivation	[12]
BrS/LQT	*SCN5A* p.(1798insAsp)	PC	Reduced I_Na_ peak current, persistent I_Na_, reduced Vmax, prolonged APD90	[13]
BrS	*SCN5A* p.(Arg620His)+ p.(Arg811His) *SCN5A* (c. 4190delA)	PC, CI	Reductions in I_Na_ and Vmax of AP, increased burden of triggered activity, abnormal calcium transients and beating interval variation	[14]
	*CACNB2* p.(Ser142Phe)	PC, CI	Reduction in peak I_Ca-L_, acceleration recovery of inactivation and altered voltage dependent inactivation, reduced APA and Vmax, reduced protein expression of the *CACNB2* gene, increased arrhythmia-like events, suppression of arrhythmic events by quinidine and bisoprolol	[15]
	*SCN5A* p.(Val1405Met) *SCNB1* p.(Ala197Val)	PC, CI	Reduction in peak I_Na_ density, reduced APA and V_max_, prolonged AP, more proarrhythmic events (EAD, DAD-like events), reduced Nav1.5 protein expression	[16]
	*SCN5A* p.(Ser1812X)	PC, IF, MEA	Reduced I_Na_ and a delayed sodium channel activation, slowed AP upstroke velocity, reduced FP and CV, enhanced I_to_ and an augmented I_Ca-L_ window current, reduced Na_V_1.5 protein expression	[17]
SQTS	*KCNH2* p.(Asn588Lys)	PC, IF, CI	Shortening APD, Increased I_Kr_ tail current, arrhythmic events, increased hERG expression, re-entrant arrhythmias	[18,19]
	*KCNH2* p.(Thr618Ile)	PC, WB	Increased I_kr_, shortened APD, beat-to-beat variability, increased membrane expression	[20]
CPVT	*RYR2* p.(Phe2483Ileu)	PC, MEA, CI	Arrhythmias, DAD, forskolin can rescue these phenotypes	[21]
	*RYR2* p.(Phe2483Ile)	CI	Longer Ca^2+^ sparks, higher diastolic Ca^2+^ levels, irregular beating, SR calcium leak and lower load levels	[22]
	*RYR2* p.(Asp3638Ala)	AFM, CI, PC,	Higher beat rate, diastolic SR Ca^2+^ leak, weaker force contraction during stress, APD, Vmax and APA decreased during stress	[23]
	*RYR2* p.(Arg176Gln)	CI	Aberrant diastolic SR Ca^2+^ release, EAD	[24]
	*RYR2* p.(Gln4201Arg) p.(Arg420Gln) p.(Phe2483Ile)	PC, CI, qPCR, WB	p.(Gln4201Arg): decrease mRNA levels RYR2, protein similar, All mutants: longer sparks p.(Arg420Gln): lower spark frequency	[25]
	*RYR2* p.(Phe13Leu) p.(Leu14Pro) p.(Arg15Pro) p.(Arg176Gln)	CI, WB, qPCR, MEA LEAP	Increased Ca^2+^ amplitude and upstroke velocity, decrease in calcium transient duration, irregular beating, decreased beat rate	[26]
	*CASQ2* p.(Asp307His)	PC, CI, EM	DADs, oscillatory arrhythmic, after-contractions and diastolic [Ca^2+^]_i_ rise, less organised myofibrils, enlarged SR cisternae and reduced number of caveolae	[27]
ACM	*PKP2* p.(Leu614Pro)	PC, CI, qPCR, IF	Reduction in rate of spontaneous cell contraction and amplitude under nifedipine, reduced expression plakophilin2 and plakoglobin	[28]
	*DSG2* p.(Gly638Arg)	IF, PC, CI, qPCR	Lower APA and Vmax, decreased peak I_Na_, I_NCX_, I_to_, I_SK_, and I_KATP_, increased I_Kr_, more arrhythmogenic events	[29]
	*DSG2* p.(Gly638Arg)	PC, WB, qPCR	Upregulation of SK4 and NDPK-B, enhanced SK4 channel currents, pacemaker activity and more arrhythmic events	[30]

Adapted and updated from Garg et al. (2018) and Pan et al. (2021) [9,10]. PC: patch clamp; IF: immunofluorescence; MEA: Multi electrode array; WB: Western Blot; CI: Calcium imaging; AFM: atomic force microscopy; AP: action potential; I_Ks_: slow delayed rectifier K+ current; ER: endoplasmic reticulum; APD50-90: Action potential duration at 50–90% of repolarisation; EAD: early after depolarisation; I_Kr_: rapid delayed rectifier K+ current; I_Ca-L_: L-type calcium current; APA: action potential amplitude; Vmax: maximum rate of rise of the action potential; I_Na_: sodium current; DAD: delayed after repolarisation; FP: field potential; CV: conduction velocity; I_to_: transient outward current; SR: sarcoplasmic reticulum; EM: electron microscope.

**Table 2 biomedicines-11-00334-t002:** Overview of published 3D iPSC-CM arrhythmia models.

3D Model	Disease/Gene/ Variant	Cellular Phenotype	Ref.
Cardiac Microtissue	ACM *PKP2* (c.2013delC, p.(Lys672ArgfsX12))	Lower Cx43 expression and arrhythmic behaviour of ACM cMT consisting of control CM and EC and ACM cardiac fibroblasts	[50]
Cardiac Microtissue	LQTS *KCNQ1* p.(Arg594Gln)	Prolonged field potential duration (FPD), β-adrenergic stimulation shortened the RR interval and decreased the FPD	[52]
Engineered heart tissue	LQTS *KCNH2* p.(Ala614Val)	APD prolongation (via ArcLight), re-entrant arrhythmic activity after I_Kr_ blocking with dofetilide	[53]
Engineered heart tissue	CPVT *CASQ2* p.(Asp307His)	More [Ca^2+^]_i_ transient abnormalities and arrhythmias compared to control EHT but less than single cell CPVT iPSC-CM	[53]

**Table 3 biomedicines-11-00334-t003:** Overview of published drug testing in iPSC-CM arrhythmia models.

Drug	Mode of Action	Disease Gene Mutation	Effect on Phenotype	Ref.
Mexiletine analogues	Class 1B antiarrhythmic drug, inhibits I_Na_	LQT *SCN5A* p.(Phe1473Cys), p.(Asn406Lys)	Mexiletine: I_NaL_ inhibition and APD shortening at lower dose but modest prolongation at higher dose and proarrhythmic response Analogues ‘MexA2′ and ‘MexA5′: more potent and selective for I_NaL_ over I_NaP_ and I_Kr_, Shortening of APD and suppression of arrhythmia Analogues ’13, 14, 25′: shortening of APD and no EADs	[76,77]
Verapamil, Lidocaine	Calcium channel blocker Sodium channel blocker	LQT *KCNQ1* p.(Gly219Glu)/ *TRPM4* p.(Thr160Met)	Reduction in APD	[78]
Telmisartan, GW0742	Agonists of the PPARδ pathway, stabilise the active PKA-phosphorylated state of hERG	LQT *KCNH2* p.(Ala561Thr)	Reduction in APD50, APD90 and triangularisation	[79]
NS1643	Change the voltage dependence of inactivation of hERG	LQT *KCNH2* p.(Ala561Thr)	Reductions in APD50, APD90 and triangularisation	[79]
Lumacaftor	Trafficking chaperone during protein folding	LQT *KCNH2* Trafficking p.(Ala561Val), (IVS9-28A/G), p.(Asn633Ser), p.(Arg685Pro), p.(Gly604Ser) Synthesis p.(Ser428X), p.(Arg366X)	*Trafficking variants* Increased membrane localisation, reduced cFPD and APD90, increase in I_Kr_ current densities, reduced calcium transient irregularities and frequency p.(Gly604Ser): increased membrane expression, no effect on APD90 *Other variants* Reduced calcium transient irregularities and frequency, no effect on cFPD	[80,81]
ICA-105574	Type II I_Kr_ activator (impairs transition to the inactivated state)	LQT *KCNH2* p.(Thr983Ile), p.( Ala422Thr)	Increased I_Kr_, shortening APD/cFPD in patient and control, shortened calcium transient, at higher concentrations (10–30 µM): cessation of the spontaneous calcium transients	[11,82]
Ajmaline	Class IA anti-arrhythmic drug inhibits I_Na_, I_to_ or I_Kr_	BrS Unknown mutation	No difference between patient and control	[83]
		BrS *SCN10A* p.(Arg1268Gln)/p.(Arg1250Gln)	Prolonged APD50 and APD90, reduced APA and Vmax	[84]
		BrS *SCN1B* p.(Leu210Pro)/ p.(Pro213Thr)	Reduced APA and Vmax	[85]
Cilostazol, Milrinone	Phosphodiesterase III inhibitors, increase I_Ca_ and suppress I_to_	BrS *SCN5A* p.(Ser1812X)	Reduction in I_to_, decreased arrhythmic beating, no EAD- or EAD-triggered activities	[17]
Bisoprolol	Beta blocker	BrS *CACNB2* p.(Ser142Phe)	Reduced arrhythmic events and reduced variation in the beat-to-beat interval time at 30 nM	[15]
Quinidine	Class I antiarrhythmic agent, blocking I_to_	BrS *CACNB2* p.(Ser142Phe)	Reduced arrhythmic events	[15]
		BrS *SCN5A* p.(Val1405Met) *SCN1B* p.(Ala197Val)	Elimination of arrhythmic events (EAD, DAD), Vmax, APA, and RMP reduced in control and patients’ groups	[16]
		SQT *KCNH2* p.(Thr618Ile)	Prolonged APD	[20]
		SQT *KCNH2* p.(Asn588Lys)	Reduced Vmax, prolonged APD, elimination of arrhythmic events	[18]
Toxin BmKKx2	Selective I_Kr_ blocker	SQT *KCNH2* p.(Thr618Ile)	Prolonged APD	[20]
Ivabradine, Ajmaline, Mexiletine	Inhibitor of the pacemaker funny current Class IA anti-arrhythmic drug, inhibits I_Na_, I_to_ or I_Kr_ Class 1B antiarrhythmic drug	SQT *KCNH2* p.(Asn588Lys)	Prolonged APD90, reduced number of arrhythmic events	[86]
MiCUps (efsevin, kaempferol, ezetimibe, disulfiram)	Mitochondrial Ca^2+^ uptake enhancers	CPVT *RYR2* p.(Ser406Leu)	Reduced number of cells displaying Ca^2+^ waves and reduced frequency of Ca^2+^ waves	[87]
		CPVT unknown mutation	Reduced Ca^2+^ waves	[88]
Autocamtide-2-related inhibitory peptide (AIP)	Ca^2+^/calmodulin-dependent protein kinase II (CaMKII) inhibitory peptide	CPVT *RYR2* p.(Ser404Arg)/p.(Asn658Ser), p.(Gly3946Ser)/p.(Gly1885Glu)	Reduced abnormal Ca^2+^ transients, reduced frequency of Ca^2+^ sparks, restored regular and spontaneous Ca^2+^ transients	[89]
Tetracaine derivative EL20	Targeted inhibition of RyR2	CPVT *RYR2* p.(Arg176Gln)	Reduced the Ca^2+^ spark frequency, prevented pacing-evoked Ca^2+^ oscillations	[24]
Nadolol, Flecainide	Non-selective beta blocker Class IC anti-arrhythmic agent inhibits I_Na_ and I_Kr_	CPVT *RYR2* p.(Phe13Leu), p.(Leu14Pro), p.(Arg15Pro), p.(Arg176Gln)	Reduced Ca^2+^ transient amplitude, reduced spontaneous Ca^2+^ release, reduced Ca^2+^ sparking activity, decreased irregularities in beat period and spontaneous beat rate	[26]

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
