# Peer review of "iPSC-Derived Cardiomyocytes in Inherited Cardiac Arrhythmias: Pathomechanistic Discovery and Drug Development"

_biomedicines, 2023, doi:10.3390/biomedicines11020334_

Round 1

Reviewer 1 Report

This is a very well-written and comprehensive review of the use of induced pluripotent stem cells to model inherited cardiac arrhythmias. It covers all pertinent and important grounds and I would wish to endorse its publication in its current form.

Author Response

Reviewer 1

This is a very well-written and comprehensive review of the use of induced pluripotent stem cells to model inherited cardiac arrhythmias. It covers all pertinent and important grounds and I would wish to endorse its publication in its current form.

Dear Reviewer,

We would like to thank you very much for your positive evaluation.

Reviewer 2 Report

Congratulations for an excellent review on this important topic for future personalized medicine approach.

Comments:

1. Please elaborate relevant clinical trials of iPSC-CM in arrhythmias.

2. Discuss the ethical concerns of the use of iPSC

3. How far we are for applying this novel technology in clinical practice?

Author Response

Reviewer 2

Congratulations for an excellent review on this important topic for future personalized medicine approach.

Dear Reviewer,

We would like to thank you very much for your positive evaluation and your useful questions and comments. You can find our replies to your comments below.

Comments:

  1. Please elaborate relevant clinical trials of iPSC-CM in arrhythmias.

Currently, there are no clinical trials using iPSC-CMs to treat arrhythmias. A few clinical trials are being performed using iPSC-CM to implant in ischemic heart failure patients (e.g. NCT04396899, NCT04945018, NCT05566600), but this is outside the scope of this article.

Some clinical trials in a dish have been performed and we mentioned them in the discussion starting at line 442. ‘As a proof-of-concept of this more ‘personalized’ drug testing, a few ‘clinical trials in a dish’ have been performed where healthy control individuals and their iPSC-CMs were challenged with known QTprolonging drugs to compare the effect on the in vitro model to the in vivo situation. Using sotalol, a correlation was found between the in vivo QT inter-val and in vitro FPD results [95]. One study also found such a positive correlation for moxifloxacin [96] while another did not find a correlation between the APD response slopes and clinical QT response to moxifloxacin or dofetilide [97]’

  1. Discuss the ethical concerns of the use of iPSC

The use of iPSCs holds less ethical concerns compared to embryonic stem cells derived from human embryos, as we now mention on line 26: ‘and less ethical concerns compared to embryonal stem cells (ESC)’
Nevertheless, the use of human cells containing genetic material, confidential personal information, potential genetic manipulation of the cells, and intellectual property and patents are ethical concerns (Moradi et al, doi: 10.1186/s13287-019-1455-y) and should be clearly outlined and discussed with the individuals donating their cells. Hence, a comprehensive informed consent is required when using these cells in research or drug testing settings. We now added this on line 447. ‘Using (subject-specific) iPSCs for research and drug testing also requires the use of a comprehensive informed consent explaining future use of created iPSCs and derivatives’.

  1. How far we are for applying this novel technology in clinical practice?

In a pharmaceutical setting, cardiotoxicity testing on iPSC-CM is already implemented. Drug testing on patient-specific iPSC-CMs followed by drug treatment in these patients is potentially available in the near future, though will be expensive. iPSC-CM cell therapy to treat arrhythmia patients is also a future possible application but because of several hurdles such as the cells’ arrhythmogenic potential and the high cost, lessons will need to be learned first from the current clinical trials for heart failure. 

For changes in text, see also answer from reviewer 3.

Reviewer 3 Report

This is an extremely interesting and well-written review by our Belgian colleagues on iPSCs and the transition from 2D to 3D models, after having excellently explained the terms of the problem with regard to the main and most dangerous cardiac arrhythmias. 

The paper is certainly well written and optimally presented; one can see that it was written by colleagues who were extremely knowledgeable about the problem, and that it was proofread and reworked by people trained in paper writing. Last but not least, to be a comprehensive review, I have not defected any plagiarism, not even the slightest, throughout the text.

What I would have expected, however, is at least a very brief 'critical' consideration of what might be the limiting aspects of the use of iPSCs, both at the clinical level (tumorigenicity, heterogeneity, and immunogenicity) and - given the target of the paper - especially at the biological level. Heterogeneity has been reported as a limitation in iPSC which to a great extent is an outcome of genetics and differences in gene expression. Some studies have also noted variability and reproducibility of results across multiple laboratories as a drawback; different reprogramming methods, chromosomal instability, can impact the phenotype of the generated iPSC. Many protocols described and published today to a great extent result in formation of immature iPSCs and the existing advancements for inducing phenotype-specific maturation has been recorded to involve a few months. Further, the need for broad range of iPSCs with respect to disease as well as cohort used for iPSC generation is a need. If colleagues could include a more critical part, albeit brief, I think the paper would be complete. 

Author Response

Reviewer 3

This is an extremely interesting and well-written review by our Belgian colleagues on iPSCs and the transition from 2D to 3D models, after having excellently explained the terms of the problem with regard to the main and most dangerous cardiac arrhythmias. 

The paper is certainly well written and optimally presented; one can see that it was written by colleagues who were extremely knowledgeable about the problem, and that it was proofread and reworked by people trained in paper writing. Last but not least, to be a comprehensive review, I have not defected any plagiarism, not even the slightest, throughout the text.

What I would have expected, however, is at least a very brief 'critical' consideration of what might be the limiting aspects of the use of iPSCs, both at the clinical level (tumorigenicity, heterogeneity, and immunogenicity) and - given the target of the paper - especially at the biological level. Heterogeneity has been reported as a limitation in iPSC which to a great extent is an outcome of genetics and differences in gene expression. Some studies have also noted variability and reproducibility of results across multiple laboratories as a drawback; different reprogramming methods, chromosomal instability, can impact the phenotype of the generated iPSC. Many protocols described and published today to a great extent result in formation of immature iPSCs and the existing advancements for inducing phenotype-specific maturation has been recorded to involve a few months. Further, the need for broad range of iPSCs with respect to disease as well as cohort used for iPSC generation is a need. If colleagues could include a more critical part, albeit brief, I think the paper would be complete. 

Dear Reviewer,

We would like to thank you very much for your positive evaluation and your useful questions and comments. You can find our replies to your comments below.

We acknowledge the limitations of the use of iPSC-CM you state and thought some of these were mentioned throughout our text, we have now included a brief section in our discussion starting at line 477. ‘Despite the immense progress that has been made in iPSC-CM generation and application potential, some limiting factors such as immaturity, genetic and phenotypic heterogeneity and variability still have an impact on their usability and should be kept in mind when translating the results in vivo [98]. For clinical application in regenerative medicine the arrhythmogenic potential, immunogenicity, tumorigenicity and heterogeneity of the iPSC-CMs should be taken into consideration.’